# Morphological peculiarities of the DNA-protein complexes in starved *Escherichia coli* cells

**Natalia Loiko[1,2], Yana Danilova[3], Andrey Moiseenko[1,3], Vladislav Kovalenko[1], Ksenia Tereshkina[1], Maria Tutukina[4,5], Galina El-Registan[2], Olga Sokolova[3,6]\*, Yurii Krupyanskii****[1]\***

**1** Department of Structure of Matter, Semenov Federal Research Center of Chemical Physics, RAS, Moscow, Russia, **2** Department of Microbiology, Federal Research Center 'Fundamentals of Biotechnology' RAS, Moscow, Russia, **3** Department of Biology, Lomonosov Moscow State University, Faculty of Biology, Moscow, Russia, **4** Center of Life Sciences, Skolkovo Institute of Science and Technology, Moscow, Russia, **5** Federal Research Center "Pushchino Scientific Center of Biological Sciences RAS", Pushchino, Russia, **6** Shenzhen MSU-BIT University, Shenzhen, China

\* sokolova@mail.bio.msu.ru (OS); yuriifkru@gmail.com (YK)

**Data Availability Statement:** All relevant data are within the manuscript and its Supporting Information files.

## Abstract

One of the adaptive strategies for the constantly changing conditions of the environment utilized in bacterial cells involves the condensation of DNA in complex with the DNA-binding protein, Dps. With the use of electron microscopy and electron tomography, we observed several morphologically different types of DNA condensation in dormant *Escherichia coli* cells, namely: *nanocrystalline*, *liquid crystalline*, and the *folded nucleosome-like*. We confirmed the presence of both Dps and DNA in all of the ordered structures using EDX analysis. The comparison of EDX spectra obtained for the three different ordered structures revealed that in *nanocrystalline* formation the majority of the Dps protein is tightly bound to nucleoid DNA. The *dps*-null cells contained only one type of condensed DNA structure, *liquid crystalline*, thus, differing from those with Dps. The results obtained here shed some light on the phenomenon of DNA condensation in dormant prokaryotic cells and on the general problem of developing a response to stress. We demonstrated that the population of dormant cells is structurally heterogeneous, allowing them to respond flexibly to environmental changes. It increases the ability of the whole bacterial population to survive under extreme stress conditions.

## Introduction

Living organisms survive in constantly changing environmental conditions, due to the universal strategies of adaptation to various stresses based on structural, biochemical, and genetic rearrangements. The universal adaptive response of microorganisms to starvation is of specific interest, because it often coincides with the resistance of pathogenic bacteria to antibiotics; this represents one of the most important medical problems in the world to date [1].

Adaptive molecular strategies ensure the ability of microorganisms to survive in environments significantly different from those, optimal for their growth, and have been the subject of active research over the past years [2–8]. Such adaptive strategy often launches the increasing

**Funding:** K.T., V.K., Y.F.K., N.L., A.M.- support from The Ministry of Science and Higher Education of the Russian Federation (state assignments #0082-2019-0015), N.L. , G.I.E. - support from The Ministry of Science and Higher Education of the Russian Federation (state assignments #AAAA-A19-119021490112-1)., OSS - supported by Russian Science Foundation (#19-74-30003), Analytical electron microscopy and dual-axis tomography were performed at the User Facility Center 'Electron microscopy in life sciences' of Moscow State University (Unique equipment setup '3D-EMC' of Moscow State University, supported by The Ministry of Science and Higher Education of the Russian Federation, identifier RFMEFI61919X0014).

**Competing interests:** The authors have declared that no competing interests exist.

synthesis (up to 150–200 thousand copies) of a ferritin-like protein Dps (DNA-binding protein of starved cells) [9–12]. Dps carries on a regulatory and protective role within *Escherichia coli* (*E. coli*) cells [13,14]. Its structure [15] and interactions with DNA were recently excessively studied *in vitro* [14,16,17], and *in silico* [18,19]. Increased synthesis of Dps in the *E. coli* cells occurs in the stationary phase, under starving conditions, allowing for the protection of DNA from oxidative stress, heat, acid, alkaline shock, toxic effects of heavy metals, antibiotics, UV radiation, etc. [9,16,20–22]. The protective functions of Dps are carried out through condensation of DNA into "*biocrystalline*" or "*in cellulo nanocrystalline*" structures [2–7,20,23–25].

It is considered that the bacterial nucleoid represents an intermediate engineering solution between the protein-free DNA packaging in viruses and protein-defined DNA packaging in eukaryotes [26]. In diluted solutions, the diameter of a DNA double-helix is about 2 nm, while its length may reach up to several centimeters. In a normal bacterial cell, circular DNA is located within a well-defined region, called 'nucleoid', which fills only 15% to 25% of the total volume. Under physiological conditions, DNA condensation leads to a dramatic decrease in the volume occupied by the DNA in the cytoplasm [27]. Polymer physicists often call this process a "coil-globule" transition [28]. By using 3C- and Hi-C-based techniques, it has been demonstrated that DNA spatial organization is different in each cell [29–31]. This corresponds to the heterogeneity of cells within a population, which allows for a flexible response to environmental changes and helps to survive in stressful situations [5].

Considering everything priory mentioned, we may expect to observe a somewhat different structural response to stress within different bacterial cells. For this reason, this study was devoted to the observation of morphological differences in structures, formed by condensed DNA in dormant *E. coli* cells and to the experimental investigation of the variety of DNA packaging in cells of different strains and growing conditions, under the stress of prolonged starvation.

## Materials and methods

### Bacterial strains

1. *E. coli* K-12 MG 1655. Wild type F- lambda- *ilvG- rfb*-50 *rph*-1 [32];

2. *E. coli* K-12 MG1655 Δ*dps*. Dps-null mutant [14];

3. *E. coli* Top10. Wild type [24];

4. *E. coli* Top10/pBAD-DPS. Dps over-producer mutant [14];

5. *E. coli* BL21-Gold(DE3)/pET-DPS. Dps over-producer mutant [14].

Bacterial strains 1 and 2 were obtained in the Laboratory of Functional genomics and cellular stress Institute of Cell Biophysics of Federal Research Center 'Pushchino Scientific Center of Biological Sciences RAS', Russia.

Bacterial strains 3–5 were kindly provided by Prof. Vassili N. Lasarev, Head of the Laboratory of Gene Engineering, Federal Research and Clinical Center of Physical-Chemical Medicine, Federal Medical Biological Agency, Moscow, Russia.

### Reagents used

All reagents were purchased from Sigma-Aldrich (St Louis, MO) or VWR (Solon, Ohio, USA), unless stated otherwise. LB medium (LB Broth, Miller (Luria-Bertani)) was purchased in VWR (Life Science, Lot: 18G3056107). Agar powder was obtained from Alfa Aesar GmbH and Co (Karlsruhe, Germany, Lot 10143436). Modified M9 medium contains the following (g/L):

Na$_2$HPO$_4$–6; KH$_2$PO$_4$–3; NaCl– 0.5; NH$_4$Cl– 0.2; MnSO$_4$–0.0004; MgSO$_4$–0.0025; CaCl$_2$–-
0.0002; glucose– 10; pH = 7.0.

## Preparation of genetically modified strains of *E. coli* Top10/pBAD-DPS and *E. coli* BL21-Gold(DE3)/pET-DPS

The DNA fragment containing the *dps* gene was obtained by PCR amplification from the *E. coli* K12 MG1655 DNA, using oligonucleotides dps-nde and dps-hind (http://www.uniprot.org/uniprot/P0ABT2):

dps-nde 5'-GATATGAACATATGAGTACCGCTAAATTAG;

dps-hind 5'-TATAAGCTTATTCGATGTTAGACTCGATAAAC.

Expression plasmids were obtained using two vectors with different transcription promoters.

The DNA fragment was introduced into the pETmin plasmid between the NdeI and HindIII restriction sites. This resulted in the production of a pET-DPS plasmid containing the DNA region encoding Dps under control of the T7 promoter.

The structure of the pBAD/Myc-His A recipient vector prevented introduction of the coding DNA fragment using restriction sites so that the recombinant protein would contain no additional amino acid sequences compared to natural Dps. The promoter was, therefore, deleted by cleavage at the BamHI and HindIII restriction sites. The excised region of the plasmid was then amplified on the pBAD/Myc-His A template, using the PB-F and PB-dpsR oligonucleotides. The Dps-encoding DNA fragment was obtained by PCR amplification of the DNA from *E. coli* K12 MG1655, using the PB-dpsF and dps-hind oligonucleotides. DNA fragments were purified by preparative electrophoresis in agarose gel, pooled, and amplified using the PB-F and PB-dpsR oligonucleotides. The DNA fragment was then introduced into the pBAD/Myc-His A plasmid, using the BamHI and HindIII restriction sites. The resultant pBAD-DPS plasmid contained the Dps-encoding DNA region under the control of an *E. coli* arabinose operon promoter.

## Dps overproduction in *E. coli* cells

*E. coli* strain BL21-Gold was transformed with the pET-DPS plasmid. *E. coli* strain Top10 was transformed with the pBAD-DPS plasmid. For protein expression, the medium (LB, Amp 150 mg/L, 10 mM lactose for BL21-Gold or 6.7 mM arabinose for Top10) was inoculated with a single bacterial colony and incubated with shaking for 16–18 hrs at 37°C. The cells were collected by centrifugation (5000 g, 15 min), resuspended in water, and homogenized with ultrasound. The lysate was centrifuged for 10 min at 13000 g. After the removal of the supernatant (water-soluble fraction), the pellet was resuspended in 1% SDS, heated for 5 min at 95°C, and centrifuged. The supernatant (insoluble fraction) was collected. Protein composition of the supernatants was analyzed by SDS-PAGE, according to Laemmli [24]. The total yield of recombinant Dps were assessed as high, constituting ~50% of the total protein in the cells of producer strains. The density of the protein bands was estimated using the gel electrophoresis image analysis software GelAnalyzer.

## Construction of the *dps*-null mutant

The *dps* deletion mutant was constructed based on the wild type K-12 MG1655 using the Gene doctoring approach [33]. In brief, regions flanking the *dps* gene were inserted in the pDOC-K plasmid between the EcoRI and XhoI restriction sites, and the resulting plasmid was co-transformed into the recipient strain (wild type K-12 MG1655) together with pACBSR possessing the lambda-red recombinase system under the control of ParaBAD. Double transformants

were grown in 0,5 ml LB (Kanamycine 20 mg/L, Chloramphenicol 50 mg/L) for 2 hrs till OD650 ~ 0.2. Then, cells were spun down (3000 rpm at room temperature), washed three times with 0.1xLB to remove any residual antibiotics, resuspended in 500 ul 0.1xLB, and recombination was induced with 0.3% arabinose. Incubation was allowed for 4 hrs, then cells were centrifuged (3000 rpm at room temperature), resuspended in 200 ul 0.1xLB, spread among three plates (Kanamycine 20 mg/L, sucrose 5%) and incubated for 36 hrs at 30˚C. Recombination efficiency was controlled at all steps exactly as described in [33]. Correctness of the *dps* deletion was confirmed by direct sequencing. **The** Kanamycine cassette was then removed using pCP20 as described in [34].

## Preparation of dormant *E. coli* cells

**K-12 MG1655.**   Bacteria were grown for 24 hrs at 28˚C under shaking (140 rpm), in 250-mL flasks with 50 mL of modified M9 medium. Cells were stored at 21˚C for 7 months.

**Top10, K-12 MG1655 *Δdps*.**   Bacteria were grown for 24 hrs at 28˚C under shaking (140 rpm), in 250-mL flasks with 50 mL of LB medium. Cells were stored at 21˚C for 7 months.

**Top10/pBAD-DPS.**   Bacteria were grown for 24 hrs at 28˚C under shaking (140 rpm), in 250-mL flasks with 50 mL of either LB or modified M9 medium, supplemented with 150 μg/mL ampicillin. Dps overproduction was induced by 1 g/l of arabinose in the linear growth phase (at 3.5 hr). After that, cells were stored at 21˚C for 7 months.

**BL21-Gold(DE3)/pET-DPS.**   Bacteria were grown for 24 hrs at 28˚C under shaking (140 rpm), in 250-mL flasks with 50 mL of modified M9 medium, supplemented with 150 μg/mL ampicillin with decreased ammonium nitrogen content. Dps overproduction was induced by 10 mM of lactose in the linear growth phase (at 3.5 hr). The cells were stored for 7 months at 21˚C.

**Light microscopy** was carried out using a Zetopan light microscope (Reichert, Austria), under phase contrast.

**The number of vegetative and dormant cells** was determined from the colony forming unit (CFU) number obtained by plating diluted cell suspensions on LB agar (2%).

## Sample preparation for electron microscopy (EM)

Cells were fixed with 2% glutaraldehyde for 5 hrs and postfixed with 0.5% paraformaldehyde; washed with a 0.1 M cacodylate buffer (pH = 7.4), contrasted with a 1% $OsO_4$ solution in a cacodylate buffer (pH = 7.4), dehydrated in an increasing series of ethanol solutions, followed by dehydration with acetone, impregnated, and embedded in Epon-812 (in accordance with manufacturer's instructions). Ultrathin sections (100–200 nm thick) were cut with a diamond knife (Diatome) on an ultramicrotome Ultracut-UCT (Leica Microsystems), transferred to copper 200 mesh grids, covered with formvar (SPI, USA), and contrasted with lead citrate, according to the Reynolds established procedure [35]. For analytical electron microscopy study, contrasting was in some cases omitted.

## Transmission electron microscopy (TEM)

Ultra-thin sections were examined in transmission electron microscopes JEM1011 and JEM-2100 (Jeol, Japan) with accelerating voltages of 80 kV and 200kV, respectively, and magnification of x13000-21000. Images were recorded with Ultrascan 1000XP and ES500W CCD cameras (Gatan, USA). Tomograms were obtained from semi-thick (300–400 nm) sections using the *Jeol Tomography* software (Jeol, Japan). The tilting angle of the goniometer ranges from -60˚ to +60˚ (with a permanent step of 1 degree). A series of images were aligned by the Gatan Digital Micrograph (Gatan, USA) and then recovered with the back-projection algorithm in IMOD4.9. 3D sub-tomograms were visualized in the UCSF Chimera package [36].

**Analytical electron microscopy** was carried out on an analytical transmission electron microscope JEM-2100 (Jeol, Japan), equipped with a bright field detector for scanning transmission electron microscopy (SPEM) (Jeol, Japan), a High Angular Angle Dark Field detector (HAADF) (Gatan, USA), an X-Max 80 mm$^2$ Silicon Drift Detector (Oxford Instruments, UK), and a GIF Quantum ER energy filter (Gatan, USA). Scanning transmission EM (STEM) and TEM modes were used. The STEM probe size was 15 nm. Energy-dispersive X-ray (EDX) spectra collection and element analyses were performed in the INCA program (Oxford Instruments, UK).

## Statistical analysis

To obtain statistically reliable data and to prove the reproducibility of the experiment data, each cell line was cultivated in three independent experiment series. The type of condensed DNA structure was determined visually, after analyzing at least 20 informative micrographs of each sample, that contained around 8–25 viable cells. Thus, the number of cells analyzed ranged from 450 to 1500 (in all the experiments combined). Results are shown in Table 2. Standard deviations of the mean values were estimated on the basis of the unbiased dispersion estimate and are shown in a Table 2. Cells with an undefined DNA structure were counted for each strain/cultivated condition and are listed in the 4$^{th}$ column of Table 2.

## Results

Dormant cells are formed by bacteria depleted from nutrients for long periods of time and are intended for the long-term survival of the population (species). [4,5,25]. Here, by the dormant state, we mean precisely only one meaning of this general term: the state that bacteria adopt because of prolonged starvation. Most cells (up to 99.98%) in long-term starving populations undergo autolysis. The remaining cells develop into dormant forms, which differ significantly in structural organization from vegetative cells. The dormant state is characterized by the absence of metabolism and the ability for growth reversion. At stress conditions, additional condensation of DNA (comparing to the physiological conditions) proceeds. The mechanism of condensation of DNA in dormant bacterial cells is still obscure.

### Dps levels increase in cells that survived starvation and stress development

The levels of Dps expression in bacterial cells after 6, 24, 48 hrs, 2 and 7 months of cultivation have been tested using the SDS-PAGE (Fig 1A). Experiments demonstrated that the biomass averaged specific levels of Dps at 42 hrs of starvation increased more than twice, comparing to protein levels in stationary cells (24 hrs), and continued to grow for up to 7 months of starvation (Fig 1B). Original uncropped and unadjusted image is located in Supporting Information files as S1 Fig.

This suggests that protein synthesis in cells continues even after the termination of active growth during transition to the stationary stage, which was demonstrated previously [37]. The heterogeneity of the bacterial population, at this stage, led to differences in the amount of Dps in individual cells, but the trend remains the same–the averaged amount of Dps (normalized per cell) is increased upon prolonged starvation, despite the decrese in the number of survived cells (Table 1).

### Three types of condensed DNA-Dps structures were found in dormant *E. coli* cells

Using transmission EM and electron two-axis tomography, we visualized and analyzed three distinct types of the DNA-Dps condensation in dormant *E. coli* cells, starving up to 7 months:

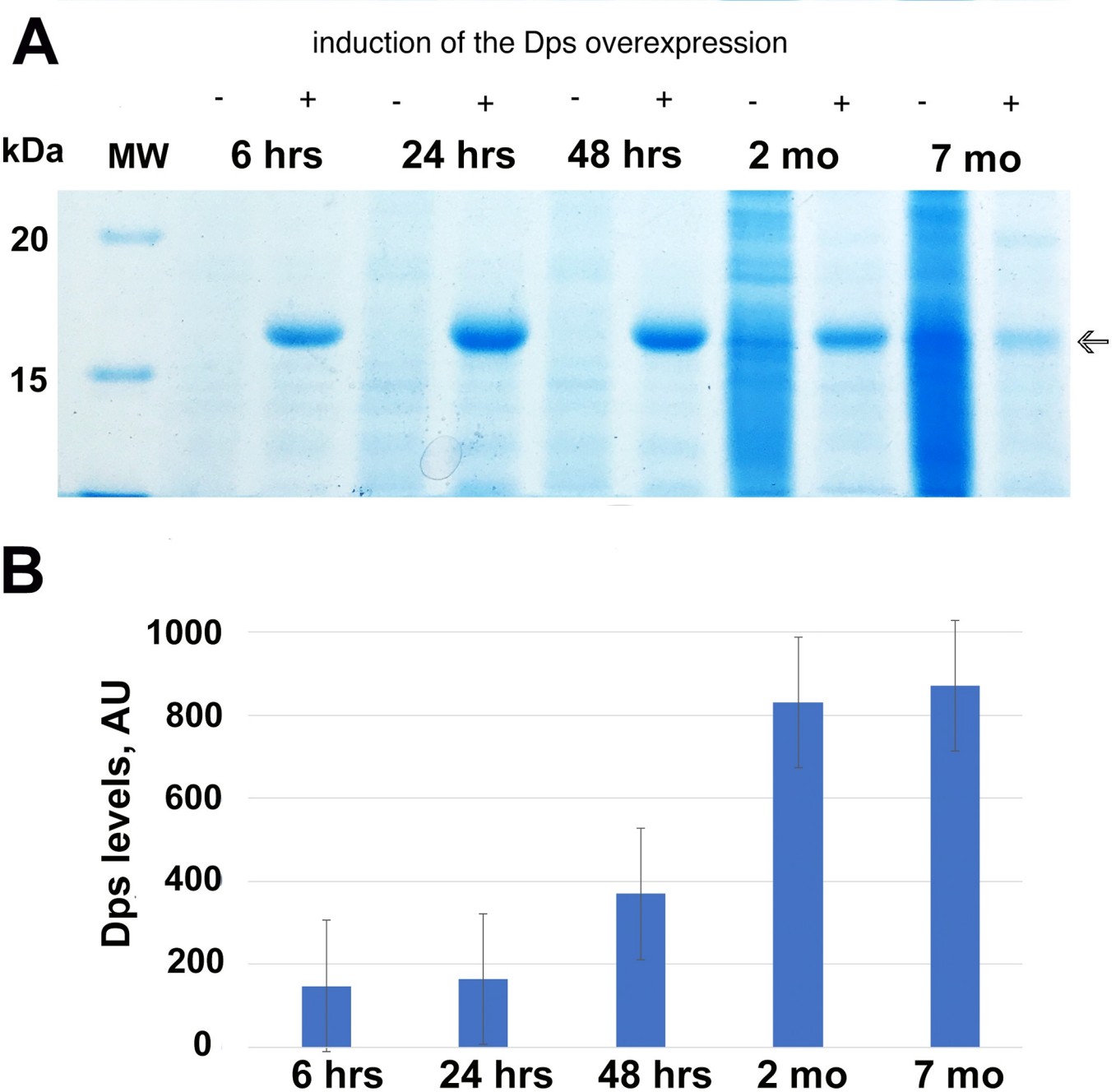

**Fig 1. Dps content in the *E. coli* cells, upon prolonged starvation.** (A) SDS-PAGE of the biomass extract. The induction of Dps overproduction was performed at the beginning of linear growth. Arrow–Dps; (B) The averaged amount of Dps (normalized per cell number) in *E. coli* Top10/pBAD-DPS population of cells at different intervals of starvation.

*nanocrystalline*, *liquid crystalline*, and, the most intriguing in our opinion, the *folded nucleosome-like* type. The first two types were extensively described previously [3,4,6,16,20,22], but the third one, to our knowledge, is novel, and has been observed here for the first time in the cells, starving for up to 7 months. *Nanocrystallines* are highly ordered intracellular structures of various sizes and shape, that often form 3D arrays, built from layers of DNA, stabilized by

**Table 1. Viable cell number decrease upon starvation.**

| The number of viable cells *E. coli* Top10/pBAD-DPS strain (CFU/ml) | | | | | | | | | |
|---|---|---|---|---|---|---|---|---|---|
| 6 hrs | | 24 hrs | | 48 hrs | | 2 months | | 7 months | |
| $1.9 \cdot 10^9$ | $4.0 \cdot 10^9$ | $1.4 \cdot 10^9$ | $3.6 \cdot 10^9$ | $7.4 \cdot 10^8$ | $1.8 \cdot 10^9$ | $5.3 \cdot 10^7$ | $2.5 \cdot 10^7$ | $2.9 \cdot 10^7$ | $6.9 \cdot 10^6$ |
| + | - | + | - | + | - | + | - | + | - |
| induction | induction | induction | induction | induction | induction | induction | induction | Induction | induction |

Dps (Fig 2A) [3,6,38]. In *liquid crystalline*, less condensed DNA is found in the central part of the cell, surrounded by a condensed cytoplasm (Fig 2B) [3]. The *folded nucleosome-like* condensate is an abundance of spherical particles with the diameter of 30–50 nm in the cytoplasm (Fig 2C). Sometimes, several types of the above-mentioned structures were co-detected (S2 Fig).

We confirmed the presence of both Dps and DNA in all ordered structures by EDX analysis (Fig 2D and 2E). This method allows to detect and map various elements on thin slices [39]. When it was possible to detect the $K_a$ peak (2.307 keV) of S, we suggested that it reflected the existence of Dps (each dodecamer of Dps contains 48 Methionines), while the $K_a$ peak (2.013

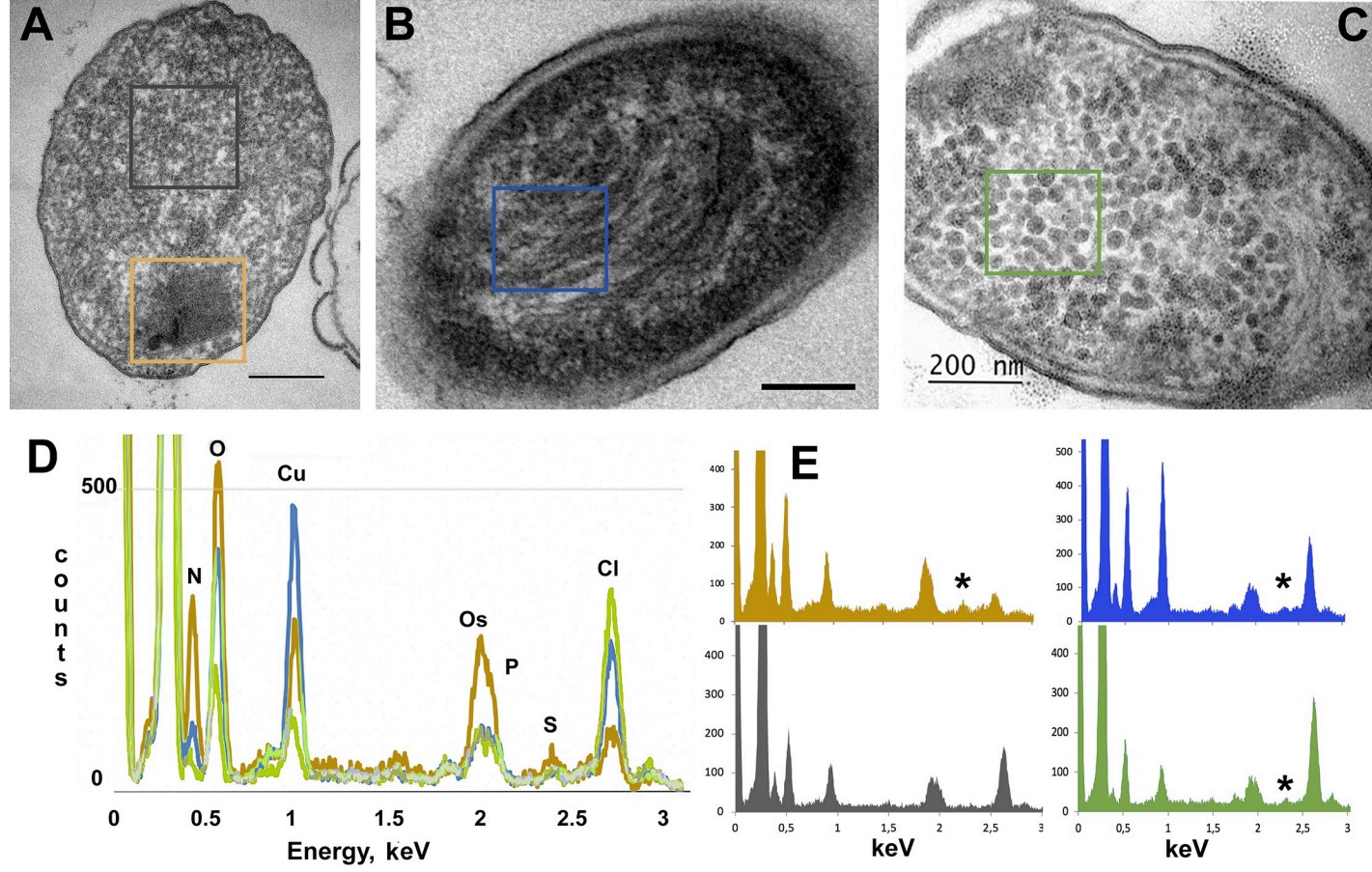

**Fig 2. Three types of condensed DNA-Dps structures were found in dormant *E. coli* cells, starving for 7 months.** (A) *nanocrystalline*; (B) *liquid crystalline*; (C) *folded nucleosome-like* type. Bar– 200 nm. Colored frames mark the specific areas, where the EDX spectra were obtained; (D) Superimposed EDX spectra from the selected areas, marked in (A-D), normalized to the C peak. Line colors reflect the corresponded frame colors in (A-D). (E) EDX spectra from the selected areas marked in (A-D), colored by the specific area. Asterisks mark positions of S peak.

keV) of P corresponded to the DNA. The areas that were subjected to EDX analysis are marked on Fig 2A–2C by colored outlines. All spectra were normalized to the C peak and superimposed to each other in Fig 2D. The pronounced Cu signal comes from the copper grids. In the reference area surrounding the ordered structure, neither S nor P have been identified (Fig 2A, dark gray rectangle; Fig 2D). It should be noted that the $K_a$ peak (2.013 keV) of P is very close to the M-line (1.914 keV) of Os, that is routinely used for fixing the cellular membranes.

Comparison of EDX spectra obtained for the three different ordered structures (Fig 2D) revealed increased levels of both P and S in *nanocrystalline*, comparing to other two formations, suggesting that in the *nanocrystal* the majority of the Dps protein is tightly bound to nucleoid DNA, forming a compact arrangement, which is in accordance with previous studies [22]. In *liquid crystalline* and *folded nucleosome-like* formations, somewhat smaller S peaks were also clearly detected (Fig 2D and 2E, asterisks), while P peaks have the same highs as the Os peaks have. This indicates that only a part of the nucleoid DNA is interacting to Dps proteins.

## Morphology of condensed DNA-Dps structures in dormant *E. coli* cells

In cells with overproduction of Dps, most of the detected structures are **nanocrystalline**. To prove this, we generated Fourier transforms from these structures (Fig 3A, insert; Fig 3C). To estimate the interlayer distance, double-axis tomography (Fig 3C) was employed. This allowed us to obtain an improved Fourier transform and, by inverse Fourier transform, an image of a DNA-Dps nanocrystal filtered from noise (Fig 3D). The distance between the centers of Dps molecules in neighboring layers was 9 nm, which is comparable with results obtained *in vitro* [38], in cryo-EM [23], and in synchrotron radiation scattering [2].

On the other hand, in cells without Dps overexpression (Fig 3B, insert), the condensed DNA-Dps structures were less ordered, which precluded us to obtain the Fourier transform from these samples. The interlayer distance in such structures dispersed from 7 to 10 nm. We did not find any *nanocrystalline* structures in dormant Dps-null cells.

***Liquid crystalline*** structure is the second common type of a DNA condensed structure that we found in dormant *E. coli* cells. The structures of this kind were demonstrated previously in starved bacteria lacking the *dps* gene [3], in viruses, and spores of various origin [26]. In some bacterial viruses, the double-stranded DNA is stored inside the capsid in the form of a spool [40,41], which can have different types of coiling leading to different types of liquid-crystalline packaging [41–43]. This packaging can change from hexagonal, to cholesteric, to isotropic at different stages of the viral life cycle.

Notably, we detected the *liquid crystalline* structures in all *E. coli* cell populations: both with the *dps* gene and lacking the *dps* gene (Dps-null mutant). In some cells, the DNA has the form of a cholesteric *liquid crystalline* order (Fig 4A and 4D). The DNA packaging in this condensed phase reduces the accessibility of DNA molecules to various damaging factors, including irradiation, oxidizing agents, and nucleases [3]. The cell, displayed on Fig 4(B), possesses the isotropic DNA condensation type [41–43], which is also characteristic for bacterial spores [26]. The DNA in the cell on Fig 4C has a nearly cholesteric order. A small amount of S (Fig 2D and 2E, asterisks) reflects the smaller Dps-to-DNA ratio (Fig 4E) in the *liquid crystalline* formations.

Of particular interest is the third type of ordered structure, found in dormant *E. coli* cells: the **folded nucleosome-like**. In all of the studied populations (except for Dps-null mutant), both with and without overproduction of Dps, the cytoplasm of 5% to 25% cells was overfilled with many round structures (Fig 5), with an average diameter of 30 nm.

Such structures were more often presented in cells that grew on a synthetic media (Table 2). The tomographic study (Fig 5) clearly demonstrated that these structures are not toroids,

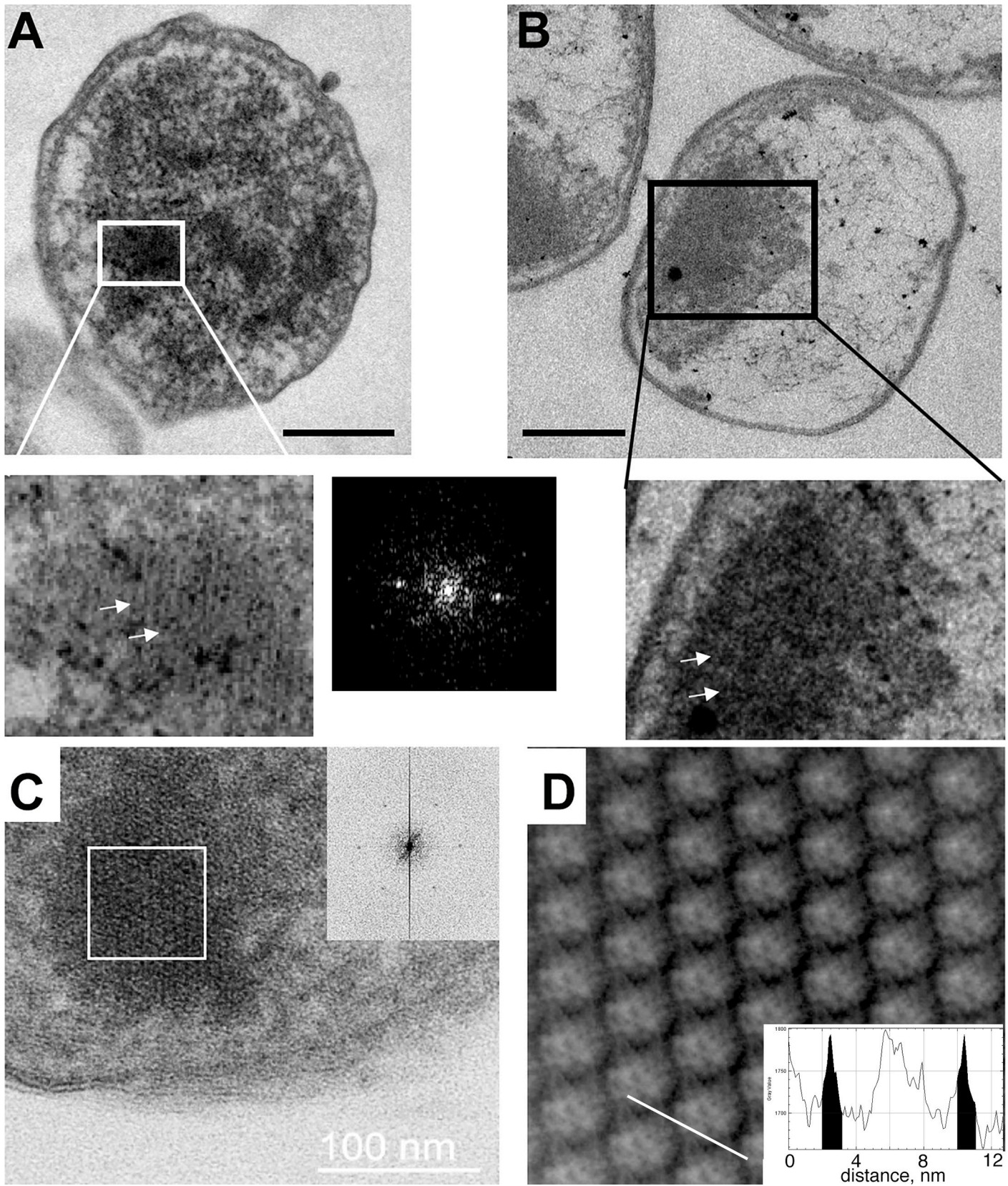

**Fig 3. Morphology of DNA-Dps nanocrystals in dormant *E. coli* cells.** (A) *E. coli* strain BL21-Gold(DE3)/pET-DPS growing on M9 media with induction of Dps overexpression in the linear growth phase, age 7 months; insert–*nanocrystalline* assembly, right–FFT from the selected in (A) area; (B) *E. coli* strain Top10/pBAD-DPS growing on LB media without inducing Dps overexpression, age 7 months. Insert–*amorphous* DNA-Dps assembly; arrows are pointing to the individual DNA-Dps layers. Bar size– 250 nm; (C) center slice through the tomogram of the *E. coli* cell (semi-thick section) with the *nanocrystalline* structure inside; insert–Fourier transform from the white-bordered area; (D) filtered DNA-Dps co-crystal. Insert–the intensity profile along the white line on the main image. Highlighted in black are densities, corresponded to the inter-layer DNA strands.

described previously [22,44,45], but represent nearly spherical formations. Remembering that the bacterial nucleoid represents an intermediate engineering solution between the protein-free DNA packaging in viruses and protein-determined DNA packaging in eukaryotes [26], we called this type of structure the '*folded nucleosome-like*'. Table 2 reflects a trend (as a percentage of the total number of cells) for the formation of a specific condensed DNA structure type for the cells of the certain strain and growth conditions after prolonged starvation.

Element analysis demonstrated that the spherical aggregates, indeed, contained S and P peaks (Fig 2D and 2E), indicating the presence of the DNA-Dps associates.

We suggested that, in bacterial cells, spherical formations of the Dps molecules (see Fig 5B and 5D) either may act similarly to histones, on which DNA is twisted (histone-like behavior) (Fig 6B), or, which is more probable, DNA arranges the Dps beads through which its string passes, forming 'beads on the string' (Fig 6A). To counteract external stress factors, these formations should be placed quite tightly on the nucleoid DNA. In addition, like in the case of protein-determined DNA packaging in eukaryotic cells where the nucleosomes are folded to form fiber loops, which then form the chromatid of a chromosome, beads on the string (or histones) may fold into a compact structure like a globule (Fig 6C). Possible schematic representation of the *folded nucleosome-like* structure generation suggested here may be seen on Fig 6. External adherent Dps molecules can additionally protect DNA.

## Dormant *E. coli* cells morphology indicates heterogeneity

The presence of specific ordered assemblies in dormant *E. coli* cells varied significantly depending on the strain and cultivation conditions (Table 2). The Dps-null dormant cells contain only *liquid crystalline* structures, in concordance with earlier studies [3]. On the other hand, in the population of dormant cells from strains Top10 (wild type) and Top10/pBAD-DPS without induction of Dps overproduction (similar to the wild type), cultivated on LB medium, the cells with the *nanocrystalline* structure were predominant (70±6%). Induction of Dps overproduction in the latter strain led to the increase of the cell number up to 77±6%. This is in accordance with earlier observations of bacterial cells starving for up to 48 hrs [2–4,16,25]. *Nanocrystalline* was the most frequently ordered structure in these cells, which consequently made them stable against the nucleases and oxidants [3,6]. Here, we observed this type of ordered structure, for the first time, in dormant cells, starving for as long as 7 months.

When the Top10/pBAD-DPS cells, without overexpression of Dps, were grown on a synthetic M9 medium (less nutrients), the number of cells with nanocrystalline structures decreased to 29±2%, but more cells with *liquid crystalline* structures appeared. In the same strain, with the overexpression of Dps, growing on synthetic M9 medium (less nutrients), a maximum number of *folded nucleosome-like* structures were detected (21±1%). This suggests that the cultivation conditions play a key role in DNA compactization. In the absence of enough nutrients, even with high concentrations of Dps (Fig 1), cells cannot form quality *nanocrystalline* structures.

Indeed, the sizes and the number of *nanocrystalline* structures in dormant cells vary, depending on the *E. coli* strain and level of Dps production. For example, in the strain BL21-Gold (DE3)/pET-DPS, with Dps induction, yet growing on M9 media, the number of

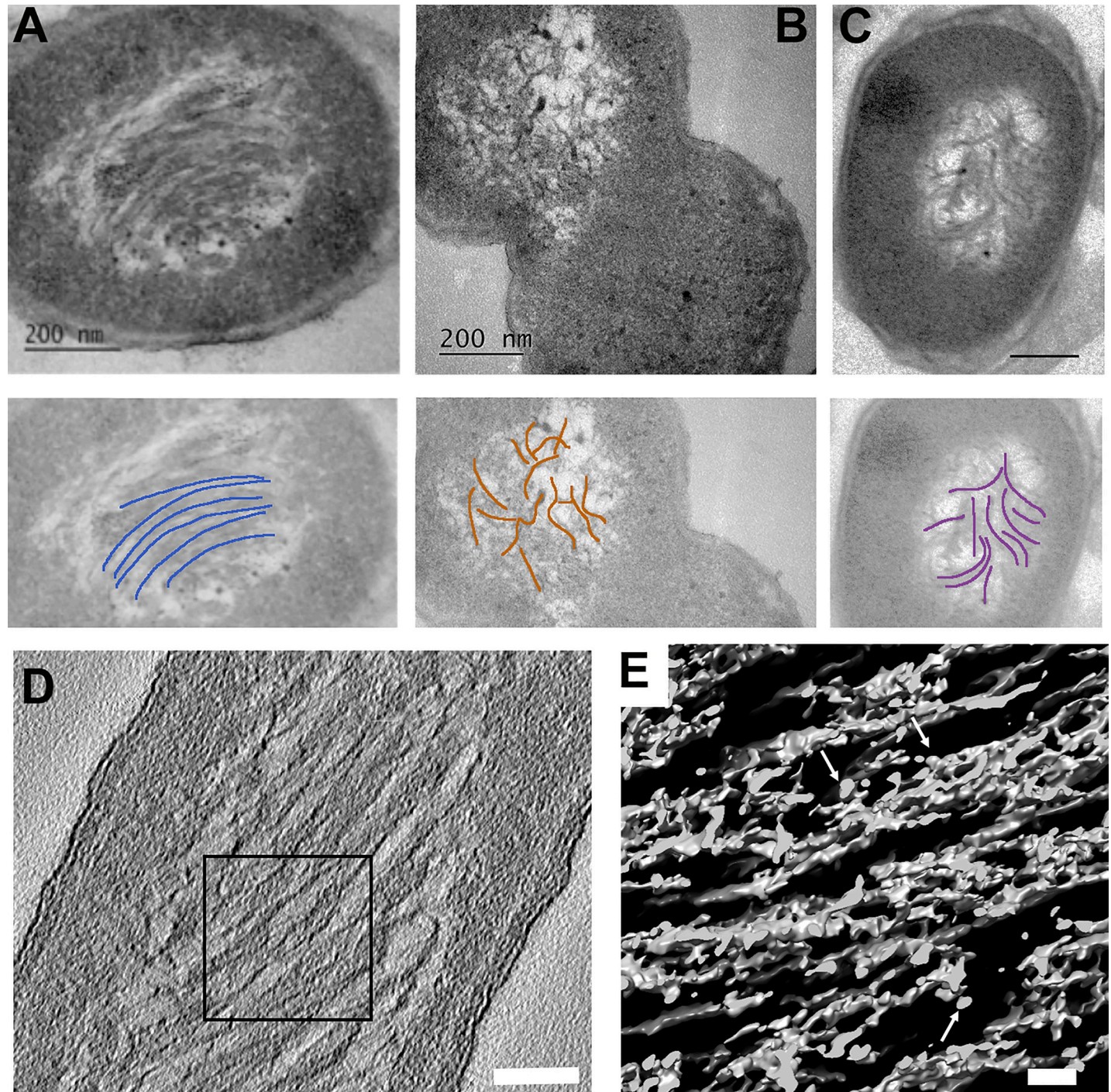

**Fig 4. Morphology of the DNA-Dps *liquid crystalline* assemblies in dormant *E. coli* cells.** (A) *E. coli* strain Top10/pBAD-DPS growing on M9 media, with induced Dps production in the linear growth phase, age 7 months; (B, C) *E. coli* strain BL21-Gold(DE3)/pET-DPS, same conditions as in (A); below, each micrograph–corresponded schematic of DNA strand distribution (blue–cholesteric order, orange–isotropic order, purple–nearly cholesteric order). (D) Central section through the tomogram of an *E. coli* cell with a cholesteric *liquid crystalline* structure. Bar –100 nm. (E) 3D representation of the *liquid crystal*, marked on (D) with a black frame. White arrows are pointing to the Dps. Bar– 30 nm.

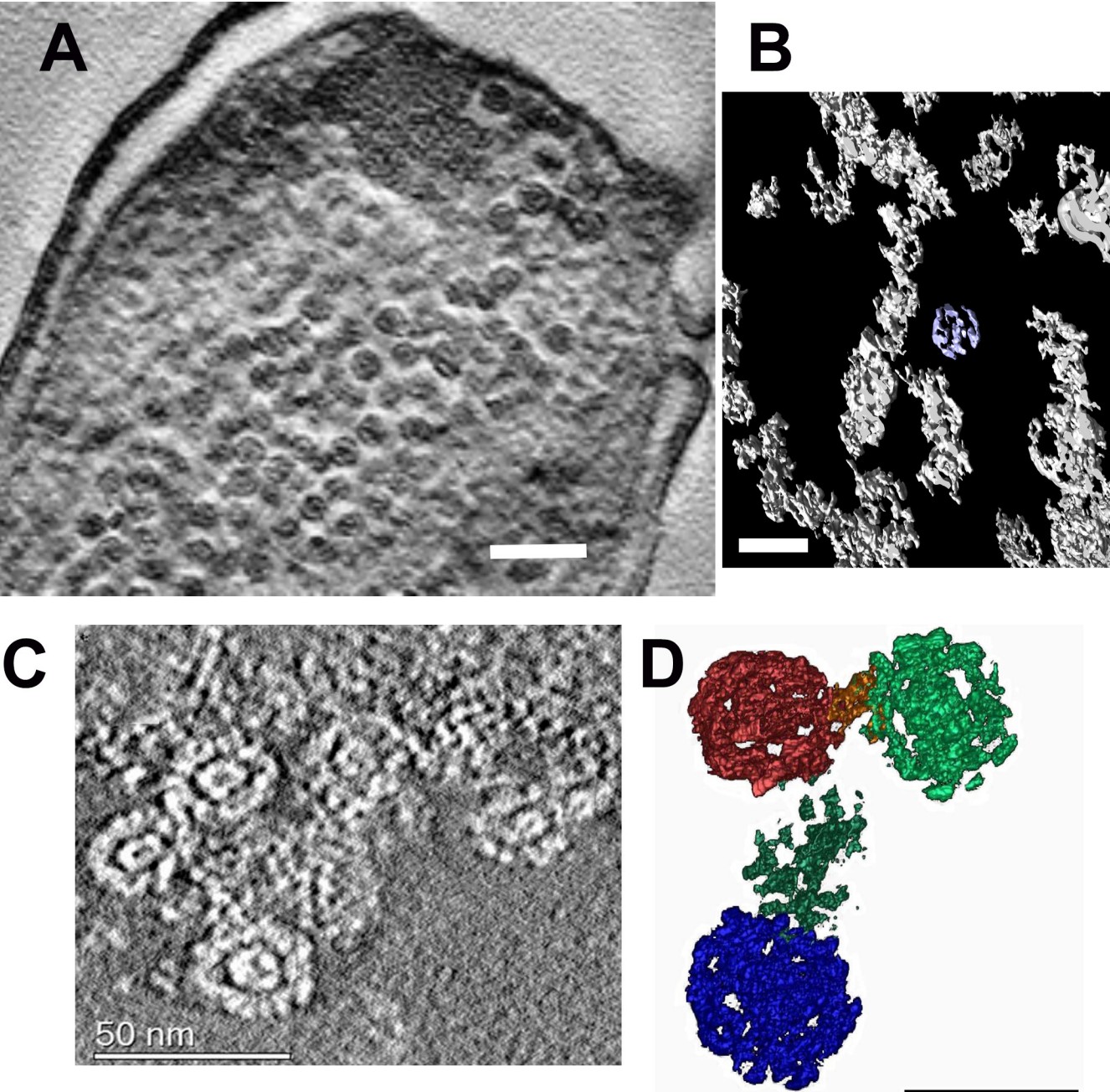

**Fig 5. Morphology of the DNA-Dps folded *nucleosome-like* structure in dormant *E. coli* cells.** (A) Tomogram of an *E. coli* cell, strain BL21-Gold(DE3)/pET-DPS growing on M9 media, with induced Dps production in the linear growth phase, age 7 months. Bar– 100 nm; (B) 3D subtomogram reconstruction of corrsponded Dps spherical associates. Bar– 50 nm; (C) Tomogram of *E. coli* cell, strain Top10/pBAD-DPS, growing on M9 media, without induction of Dps production, age 7 months. Bar size– 50 nm; (D) 3D subtomogram reconstruction of corresponded Dps spherical associates. Bar size– 30 nm.

nanocrystals usually varied from 5 to 10 per cell; with sizes of approximately 40–80 nm (Fig 3A). In the Top10/pBAD-DPS cells, without Dps induction, grown on an LB medium and aged 7 months, one large crystal (300–400 nm size) often occupied a large part of one cell (Fig 3B). The number of cells bearing *liquid crystalline* structures in populations of dormant *E. coli* cells which expressed Dps, ranged from ~8% up to ~49% (Table 2), with the majority being observed in those cells growing on M9 synthetic medium.

**Table 2. Variety of condensed DNA-Dps types in studied *E. coli* dormant cells.**

| Sample description | | | Types of condensed structure, % of cells in the population | | | |
|---|---|---|---|---|---|---|
| *E. coli* strain | Cultivation conditions | | *# nanocrystalline* | *# liquid crystalline* | *# folded nucleosome-like* | *Undefined structure* |
| | Growth medium | Dps overexpression | | | | |
| K-12 MG1655 Δ*dps* | LB | - | 0 | **73 ± 6** | 0 | 27 ± 3 |
| K-12 MG1655 | M9 | - | **47 ± 4** | 31 ± 2 | 13 ± 2 | 9 ± 1 |
| Top 10 | LB | - | **69 ± 6** | 20 ± 2 | 4 ± 1 | 7 ± 1 |
| Top10/pBAD-DPS | LB | - | **71 ± 6** | 16 ± 1 | 5 ± 1 | 8 ± 1 |
| | M9 | - | 26 ± 2 | **50 ± 4** | 14 ± 1 | 10 ± 1 |
| | LB | + | **78 ± 6** | 8 ±1 | 5 ± 1 | 9 ± 1 |
| | M9 | + | 32 ± 2 | **36 ± 3** | 21 ± 1 | 11 ± 1 |
| BL21-Gold(DE3)/pET-DPS | M9 | + | **50 ± 4** | 23 ± 2 | 17 ± 1 | 10 ± 1 |

## Discussion

Several decades of research established, not in a high-resolution quality yet, but nevertheless, the very beautiful and sophisticated architecture of the *E. coli* nucleoid in the active growing phase [46]. Such architecture may only exist, due to the dynamic order that characterizes actively growing bacteria [22]. In the stationary phase caused by nutrient depletion, energy production processes become inefficient. Bacteria in the prolonged stationary phase use another, energy-independent mechanism for maintaining order and for the protection of vital structures (DNA): the creation of stable structures, similar to those in inanimate nature.

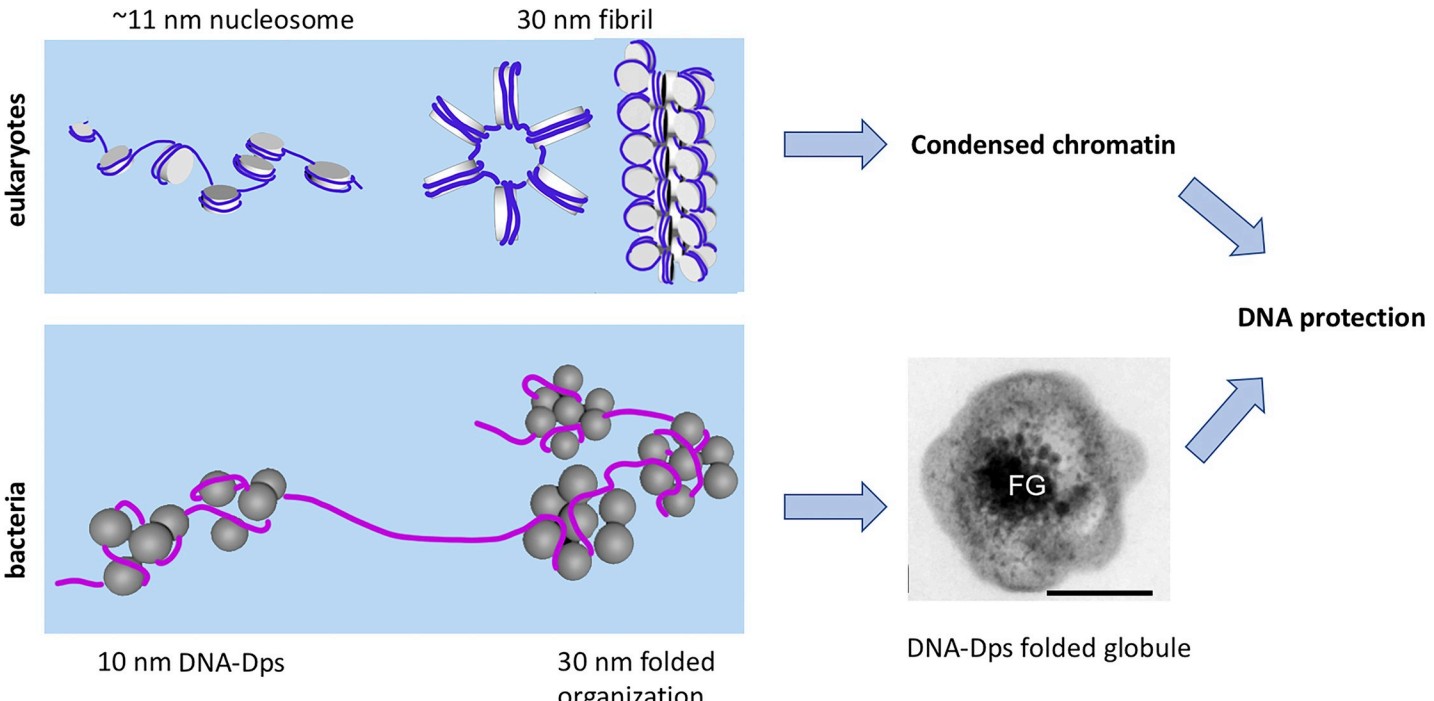

**Fig 6. The comparison of pro- and eukaryotic DNA compactization.** Top: schematic–eukaryotic nucleosomes (histone proteins wrapped with DNA) fold up into 30 nm fibrils that, in their turn, are compressed to produce fiber and chromosomes, which protect the DNA from external factors. Bottom schematic–prokaryotes do not have defined chromosomes, but the DNA is folded around the Dps molecules to form 'beads on the string' first (as in Fig 5D), these could fold into spherical aggregates 30 nm in diameter and, further, into a globule-like structure, which effectively protects the nucleoid DNA from stresses.

In this work, we observed three types of stable DNA condensation in dormant *E. coli* cells. First two: *nanocrystalline* and *liquid crystalline* structures, are typical for inanimate nature. The third one: *folded nucleosome-like*, may be the result of complex interaction and multiple folding of long DNA molecules around Dps dodecamers and their associates (Fig 6). The complex structure of a condensed high polymer like DNA may consist of various regions, each with a characteristic degree of the internal order [47] ranging continuously from some of it close to the ideal crystals, to a completely amorphous state. Earlier we studied dormant *E.coli* cells using synchrotron radiation diffraction [2,4]. Broad diffraction peaks found there were indicative of the imperfection of the DNA-Dps co-crystals. The fact that cells bearing the above-mentioned ordered forms were found in aged populations proves that they are not transitional intermediate forms, but that they are programmed into the development cycle for the purpose of implementing different survival strategies. DNA does not form a unique compact structure upon condensation. The heterogeneity of cells allows to respond flexibly to environmental changes and to survive in stressful situations.

Multiple types of DNA condensation in the same dormant *E. coli* cell increase the chances of the cells for survival and for rapid resumption of growth when conditions turn out to be favorable.

Many adverse environmental conditions, such as extreme desiccation, freezing, oxygen deficiency, factors of suspended animation (anabiosis), bacteriophages infection, etc., can affect the ability to survive, and, subsequently, the morphology and ultrastructure of the dormant forms. In further studies, the study of DNA condensation in other types of dormant cells subjected to different stress factors will help understand why cells need heterogeneity for survival. This may shed some light on the understanding of the mechanisms of bacterial resistance to antibiotics, which represents one of the most important medical problems in the world to date and will allow to solve a number of problems of bioengineering and medicine.

## Supporting information

**S1 Fig. Dps content in the *E. coli* cells, upon prolonged starvation.**
(TIF)

**S2 Fig. *E. coli* Top10/pBAD-DPS cell with several types of condensed DNA-Dps structures.**
NK–*nanocrystalline; arrow—folded nucleosome-like* condensate.
(TIF)

## Acknowledgments

Authors would like to thank Prof. Vassili N. Lasarev for providing *E. coli* strains, Evgeniy Kulikov and Alla Golomidova for help with gel electrophoresis and Lisa Trifonova for proofreading the manuscript.

## Author Contributions

**Conceptualization:** Natalia Loiko, Olga Sokolova, Yurii Krupyanskii.

**Data curation:** Natalia Loiko, Maria Tutukina.

**Investigation:** Yana Danilova, Andrey Moiseenko, Vladislav Kovalenko, Galina El-Registan.

**Software:** Vladislav Kovalenko, Ksenia Tereshkina.

**Writing – original draft:** Natalia Loiko, Olga Sokolova, Yurii Krupyanskii.

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
