## [Decision Letter · Decision Letter 0]

7 Jul 2020

PONE-D-20-08434

Morphological peculiarities of DNA-protein complexes in dormant Escherichia coli cells, subjected to prolonged starvation

PLOS ONE

Dear Dr. Yurii Krupyanskii,

Thank you for submitting your manuscript to PLOS ONE, and my personal apologies for getting back to you with a propper review. After careful consideration, we feel that it has merit but does not fully meet PLOS ONE’s publication criteria as it currently stands.

Therefore, we invite you to submit a revised version of the manuscript that convincingly addresses all the points raised during the review process and that both reviewers have constructively indicated.

We look forward to receiving your revised manuscript.

Kind regards,

Maria Gasset, Ph.D.

Academic Editor

PLOS ONE

Journal Requirements:

2.PLOS ONE now requires that authors provide the original uncropped and unadjusted images underlying all blot or gel results reported in a submission’s figures or Supporting Information files. This policy and the journal’s other requirements for blot/gel reporting and figure preparation are described in detail at https://journals.plos.org/plosone/s/figures#loc-blot-and-gel-reporting-requirements and https://journals.plos.org/plosone/s/figures#loc-preparing-figures-from-image-files. When you submit your revised manuscript, please ensure that your figures adhere fully to these guidelines and provide the original underlying images for all blot or gel data reported in your submission. See the following link for instructions on providing the original image data: https://journals.plos.org/plosone/s/figures#loc-original-images-for-blots-and-gels.

Reviewers' comments:

Reviewer's Responses to Questions

**Comments to the Author**

1. Is the manuscript technically sound, and do the data support the conclusions?

Reviewer #1: Partly

Reviewer #2: Yes

2. Has the statistical analysis been performed appropriately and rigorously? 

Reviewer #1: Yes

Reviewer #2: No

3. Have the authors made all data underlying the findings in their manuscript fully available?

Reviewer #1: Yes

Reviewer #2: Yes

4. Is the manuscript presented in an intelligible fashion and written in standard English?

Reviewer #1: Yes

Reviewer #2: Yes

5. Review Comments to the Author

Reviewer #1: The paper by Loiko et al reports an electron microscopy study of the complexes formed by the nucleoid-associated protein Dps of E.coli and DNA in cells that survive a prolonged starvation. To this end, Authors prepare strains that overproduce Dps upon induction of alternate expression systems (either arabinose-induced or T7-based) and test their viability along time in different growth media. The long-lasting cells are then inspected with various types of EM and found to contain a range of structures which in one case resemble nucleosomes.

The paper is interesting and visually appealing, but there is a number of issues that should be tackled:

1. A clarification of the biological momenclature would be very welcome. I do not understand the pecise meaning of *dormant*. Is it that they keep some membrane potential—but can they generate viable colonies? What is *vegetative* in this context (eg Table 1)? How do we know the cells observed under the microscope are alive or dead?

2. The Achilles's heel of this work is that the interesting structures observed under EM come from cells overexpressing Dps. The wt is mentioned in Table 2, but not analyzed further. It is worrisome that the distribution of the various DNA-protein complexes depends on the overexpression system. BTW, the Table lacks various controls eg strains carrying insert-less plasmids (not just the wt) with and without inducers.

3. Is Dps protein essential? It would be highly informative to inspect starved cells lacking the protein. The nucleoid has many other associated proteins which could contribute to the structures shown in the figures.

Reviewer #2: Bacterial cells have a variety of mechanisms of molecular adaptation for survival under stress conditions. One of them relates to the protein Dps (DNA-binding protein of starved cells), which acts as a protective element against starvation by interacting with bacterial DNA and promoting its condensation in supramolecular structures.

This work describes the analysis of the structural organization of the condensed DNA-Dps assemblies in E. coli cells starved for extended periods (months) by electron microscopy and electron tomography. The authors have found three types of condensed DNA-Dps structures in dormant cells, two previously reported (nanocrystalline and liquid crystalline structures). In contrast, the third one - termed folded nucleosome-like structures - is novel being more abundant in cells grown in synthetic medium. They conclude that the morphological heterogeneity of DNA condensates in dormant cells found in this study here is an additional factor contributing to the response of bacterial cells to stress conditions and environmental changes.

COMMENTS

Title: please try to shorten it.

Introduction: it supplies sufficient background information to allow the reader to understand and evaluate the results of the present study and to provide the rationale of this work.

The experimental procedures used are appropriate to ask the proposed questions. I encourage the improvement of the statistical analysis of the observations summarized in Table 2, including some statistical parameters (i.e., the standard deviation of the mean values, or equivalent) to allow the reader to evaluate if the abundance of the difference morphologies found is significant within experimental uncertainty or not.

The description of the results is again concise and clear. The figures complement well the main text.

The manuscript will improve if the authors further elaborate on the significance of the novel structures found here (namely the ones denoted as folded nucleosome-like structures). In this regard, Figure 6 would need further improvements. How these results and the interpretations of them agree (or contrast) with previously published work?

In the conclusions, could the authors define more precisely the unsettled points for future explorations? What are the principles, relationships, and generalizations shown by the results?

6. PLOS authors have the option to publish the peer review history of their article (what does this mean?). If published, this will include your full peer review and any attached files.

Reviewer #1: No

Reviewer #2: No

---

## [Author Response · Author response to Decision Letter 0]

7 Sep 2020

Journal Requirements:

2.PLOS ONE now requires that authors provide the original uncropped and unadjusted images underlying all blot or gel results reported in a submission’s figures or Supporting Information files. This policy and the journal’s other requirements for blot/gel reporting and figure preparation are described in detail at https://journals.plos.org/plosone/s/figures#loc-blot-and-gel-reporting-requirements and https://journals.plos.org/plosone/s/figures#loc-preparing-figures-from-image-files. When you submit your revised manuscript, please ensure that your figures adhere fully to these guidelines and provide the original underlying images for all blot or gel data reported in your submission. See the following link for instructions on providing the original image data: https://journals.plos.org/plosone/s/figures#loc-original-images-for-blots-and-gels.

Reviewers' comments:

Reviewer's Responses to Questions

Comments to the Author

1. Is the manuscript technically sound, and do the data support the conclusions?

Reviewer #1: Partly

Reviewer #2: Yes

2. Has the statistical analysis been performed appropriately and rigorously?

Reviewer #1: Yes

Reviewer #2: No

3. Have the authors made all data underlying the findings in their manuscript fully available?

Reviewer #1: Yes

Reviewer #2: Yes

4. Is the manuscript presented in an intelligible fashion and written in standard English?

Reviewer #1: Yes

Reviewer #2: Yes

5. Review Comments to the Author

Reviewer #1: The paper by Loiko et al reports an electron microscopy study of the complexes formed by the nucleoid-associated protein Dps of E.coli and DNA in cells that survive a prolonged starvation. To this end, Authors prepare strains that overproduce Dps upon induction of alternate expression systems (either arabinose-induced or T7-based) and test their viability along time in different growth media. The long-lasting cells are then inspected with various types of EM and found to contain a range of structures which in one case resemble nucleosomes.

The paper is interesting and visually appealing, but there is a number of issues that should be tackled:

1. A clarification of the biological momenclature would be very welcome. I do not understand the pecise meaning of *dormant*. Is it that they keep some membrane potential—but can they generate viable colonies? What is *vegetative* in this context (eg Table 1)? How do we know the cells observed under the microscope are alive or dead?

2. The Achilles's heel of this work is that the interesting structures observed under EM come from cells overexpressing Dps. The wt is mentioned in Table 2, but not analyzed further. It is worrisome that the distribution of the various DNA-protein complexes depends on the overexpression system. BTW, the Table lacks various controls eg strains carrying insert-less plasmids (not just the wt) with and without inducers.

3. Is Dps protein essential? It would be highly informative to inspect starved cells lacking the protein. The nucleoid has many other associated proteins which could contribute to the structures shown in the figures.

Reviewer #2: Bacterial cells have a variety of mechanisms of molecular adaptation for survival under stress conditions. One of them relates to the protein Dps (DNA-binding protein of starved cells), which acts as a protective element against starvation by interacting with bacterial DNA and promoting its condensation in supramolecular structures.

This work describes the analysis of the structural organization of the condensed DNA-Dps assemblies in E. coli cells starved for extended periods (months) by electron microscopy and electron tomography. The authors have found three types of condensed DNA-Dps structures in dormant cells, two previously reported (nanocrystalline and liquid crystalline structures). In contrast, the third one - termed folded nucleosome-like structures - is novel being more abundant in cells grown in synthetic medium. They conclude that the morphological heterogeneity of DNA condensates in dormant cells found in this study here is an additional factor contributing to the response of bacterial cells to stress conditions and environmental changes.

COMMENTS

Title: please try to shorten it.

Introduction: it supplies sufficient background information to allow the reader to understand and evaluate the results of the present study and to provide the rationale of this work.

The experimental procedures used are appropriate to ask the proposed questions. I encourage the improvement of the statistical analysis of the observations summarized in Table 2, including some statistical parameters (i.e., the standard deviation of the mean values, or equivalent) to allow the reader to evaluate if the abundance of the difference morphologies found is significant within experimental uncertainty or not.

The description of the results is again concise and clear. The figures complement well the main text.

The manuscript will improve if the authors further elaborate on the significance of the novel structures found here (namely the ones denoted as folded nucleosome-like structures). In this regard, Figure 6 would need further improvements. How these results and the interpretations of them agree (or contrast) with previously published work?

In the conclusions, could the authors define more precisely the unsettled points for future explorations? What are the principles, relationships, and generalizations shown by the results?

6. PLOS authors have the option to publish the peer review history of their article (what does this mean?). If published, this will include your full peer review and any attached files.

Do you want your identity to be public for this peer review? For information about this choice, including consent withdrawal, please see our Privacy Policy.

---

## [Decision Letter · Decision Letter 1]

16 Sep 2020

Morphological peculiarities of DNA-protein complexes in starved Escherichia coli cells

PONE-D-20-08434R1

Dear Dr.  Yuri Krupyanskii, 

We’re pleased to inform you that your manuscript has been judged scientifically suitable for publication and will be formally accepted for publication once it meets all outstanding technical requirements.

Kind regards,

Maria Gasset, Ph.D.

Academic Editor

PLOS ONE

Additional Editor Comments (optional):

Reviewers' comments:

Reviewer's Responses to Questions

**Comments to the Author**

1. If the authors have adequately addressed your comments raised in a previous round of review and you feel that this manuscript is now acceptable for publication, you may indicate that here to bypass the “Comments to the Author” section, enter your conflict of interest statement in the “Confidential to Editor” section, and submit your "Accept" recommendation.

Reviewer #1: All comments have been addressed

2. Is the manuscript technically sound, and do the data support the conclusions?

Reviewer #1: Yes

3. Has the statistical analysis been performed appropriately and rigorously? 

Reviewer #1: Yes

4. Have the authors made all data underlying the findings in their manuscript fully available?

Reviewer #1: Yes

5. Is the manuscript presented in an intelligible fashion and written in standard English?

Reviewer #1: Yes

6. Review Comments to the Author

Reviewer #1: (No Response)

7. PLOS authors have the option to publish the peer review history of their article (what does this mean?). If published, this will include your full peer review and any attached files.

Reviewer #1: No

---

## [Editor Report · Acceptance letter]

24 Sep 2020

PONE-D-20-08434R1 

Morphological peculiarities of DNA-protein complexes in starved Escherichia coli cells 

Dear Dr. Krupyanskii:

I'm pleased to inform you that your manuscript has been deemed suitable for publication in PLOS ONE. Congratulations! Your manuscript is now with our production department. 

Kind regards, 

on behalf of

Dr. Maria Gasset 

Academic Editor

PLOS ONE